# A year of pandemic: Levels, changes and validity of well-being data from Twitter. Evidence from ten countries

Francesco Sarracino[1], Talita Greyling[2,3], Kelsey O'Connor[1,2], Chiara Peroni[1], Stephanié Rossouw[2,3] *

**1** Research Division, Institut national de la statistique et des études économiques du Grand-Duché du Luxembourg, Luxembourg, Luxembourg, **2** School of Economics, College of Business and Economics, University of Johannesburg, Johannesburg, South Africa, **3** School of Social Science & Public Policy, Faculty of Culture & Society, Auckland University of Technology, Auckland, New Zealand

☯ These authors contributed equally to this work.
* stephanie.rossouw@aut.ac.nz

**Data Availability Statement:** All relevant data are within the manuscript and its Supporting Information files.

## Abstract

We use daily happiness scores (Gross National Happiness (GNH)) to illustrate how happiness changed throughout 2020 in ten countries across Europe and the Southern hemisphere. More frequently and regularly available than survey data, the GNH reveals how happiness sharply declined at the onset of the pandemic and lockdown, quickly recovered, and then trended downward throughout much of the year in Europe. GNH is derived by applying sentiment and emotion analysis–based on Natural Language Processing using machine learning algorithms–to Twitter posts (tweets). Using a similar approach, we generate another 11 variables: eight emotions and three new context-specific variables, in particular: trust in national institutions, sadness in relation to loneliness, and fear concerning the economy. Given the novelty of the dataset, we use multiple methods to assess validity. We also assess the correlates of GNH. The results indicate that GNH is negatively correlated with new COVID-19 cases, containment policies, and disgust and positively correlated with staying at home, surprise, and generalised trust. Altogether the analyses indicate tools based on Big Data, such as the GNH, offer relevant data that often fill information gaps and can valuably supplement traditional tools. In this case, the GNH results suggest that both the severity of the pandemic and containment policies negatively correlated with happiness.

## 1. Introduction

Improving individuals' well-being is not only a desirable outcome "per se", but it also carries wider individual and societal benefits. Happier people tend to live longer and healthier lives (Danner et al. [1], Guven and Saloumidis [2], Frijters et al. [3], Graham and Pinto [4]), have better employment outcomes (O'Connor [5]) and share creative, altruistic and problem-solving traits (Lyubomirsky et al. [6]). Happier people are more satisfied with their jobs, are more productive, cooperative and less absent (Bryson et al. [7], DiMaria et al. [8], Oswald et al. [9]).

**Funding:** This study was funded by Fonds National de la Recherche Luxembourg (grant number FNR-14878312) awarded to FS. This study was also funded by the Auckland University of Technology, awarded to SR, and by the University of Johannesburg, awarded to TG. The funders had no role in study design, data collection and analysis, decision to publish, or preparation of the manuscript.

**Competing interests:** The authors have declared that no competing interests exist.

What is more, higher levels of past and present happiness predict higher levels of compliance during COVID-19 (Krekel et al. [10]). In general, the literature shows that traumatic events alter well-being in rapid and persistent ways (Bonanno et al. [11], Kessler et al. [12], Norris et al. [13]). This, in turn, has long-lasting collateral social and economic effects (Arampatzi et al. [14]). The novel coronavirus pandemic is undoubtedly one such event.

We know that the spread of COVID-19 has deeply affected society's well-being, directly and through containment policies' socioeconomic consequences. Additionally, well-being has important societal consequences and can affect the effectiveness of responses to COVID-19. This latter point is especially relevant to policymaking, as individuals' responses are key to health policies' effectiveness and affect the successfulness of "exit" strategies to ease lockdowns and recovery plans. Timely well-being data is particularly relevant during the pandemic, as it can facilitate rapid policy responses to changing conditions.

Given the above, this study's primary aim is to use timely data to analyse happiness, and its correlates in ten countries across Europe and the Southern hemisphere during 2020. To account for the simultaneous effect of various factors on the changes in well-being over time, we use regression analysis.

Our analyses use a unique dataset, which we construct from tweets extracted in real-time at the country level. We use Natural Language Processing (machine learning) to extract the tweets' underlying sentiment and emotions. We apply algorithms to the sentiment and emotion scores and derive daily time-series data per country for happiness and eight emotions (see section 3.1.1).

We measure happiness using the Gross National Happiness index (GNH). It has been used in multiple previous studies (see section 3.2.1) (Greyling et al. [15]); however, the current study extends the coverage of the GNH to include seven European countries. Additionally, we create three new context-specific variables by filtering tweets on keywords and then analysing the emotions of these subsets of tweets, specifically, trust in institutions, sadness related to loneliness, and economic fear. We perform a significant number of validity tests, which we believe suggests GNH provides meaningful (though not perfect) indications of national happiness. We then assess the correlates of GNH in various regression analyses.

The primary benefit of the GNH and the derived Twitter dataset is its timeliness. In contrast, traditional well-being measures, such as those coming from large-scale surveys, report changes in well-being and other variables after a significant time lag. Surveys are also costly and take a considerable amount of time to administer. Indeed, the pandemic disrupted survey administration, which reduced the amount of data on well-being and its correlates. For instance, the Eurobarometer, a European Commission survey, usually provides the most recent comparable well-being series for European countries at a biannual frequency. In 2019, the survey was administered twice. However, in 2020 the survey was only administered once. Fewer surveys are particularly challenging in times of crisis as accurate information is necessary to make important and difficult choices rapidly.

Surveys with only one reference point in time also do not tell us anything about the evolution of well-being during the period between the COVID-19 waves. For example, the Eurobarometer data collected for Luxembourg in 2020 indicate that life satisfaction decreased by eight percentage points between Autumn 2019 (94%) and Summer 2020 (86%) (European Commission [16]). It is plausible that this large decrease in subjective well-being is due to the pandemic, but questions remain about the dynamics, true size, and mechanisms underlying the decrease. The true decrease might have been larger or smaller than eight percentage points because we only know the amount observed when the Eurobarometer survey was fielded. High-frequency data are needed to address these issues, particularly given policymakers' challenges during a pandemic.

We make several contributions to the literature. This is the first study to document the changes in happiness during the pandemic in ten countries across both Europe and the Southern hemisphere using timely Big Data. Second, this is the first study we are aware of using a unique method of filtering tweets on keywords to derive time-series data for the following variables: trust in institutions, economic fear, and sadness in relation to loneliness. In doing so, we also contribute to the methodological literature on applying sentiment and emotion analysis to Big Data. Third, using tweets to construct variables related to well-being is of general interest and use to decision-makers and statistical offices, partly because the methods and data are applicable beyond the current pandemic. Fourth, this is the first study to test the validity of happiness and related measures sourced from Twitter by analysing their correlation across countries and time with available survey data and other Big Data sources. Finally, the study contributes to the methodological research on developing new and timely measures of socio-economic variables using Big Data and machine learning (Brodeur et al. [17], Greyling et al. [18]).

Our results suggest GNH provides meaningful information on national happiness–GNH correlates meaningfully (though not significantly, possibly due to sample sizes) with alternative measures of well-being and ill-being from surveys and other Big Data sources. The same holds for economic fear, trust in national institutions, and generalised trust. In terms of the trend over time, we saw that a decrease in the GNH corresponded with a rise in infection and more stringent containment policies. Regression analyses indicate that changes in GNH correlate negatively with changes in new positive cases (particularly the increases) and with the expected increase in containment policy stringency. An increase in people staying at home predicts an increase in GNH. In other words, ceteris paribus, the more time spent at home, the higher was GNH. Results also indicate that economic fear, trust in national institutions and sadness related to loneliness are not significantly associated with changes in GNH. Finally, we found that the GNH correlates positively with increases in surprise and generalised trust and decreases in disgust.

The paper is organised as follows. The next section reviews the literature on the impact of COVID-19 on well-being and discusses the use of Big Data to measure well-being. Section 3 describes the data and the selected variables and outlines the methodology used to construct our Twitter dataset. Section 4 outlines the methodology followed in our regression analysis to determine the correlates of changes in the GNH. The results and discussion follow in sections 5 and 6, while the paper concludes in section 7.

## 2. Literature review

### 2.1 The impacts of COVID-19 on well-being

An extensive and interdisciplinary literature discusses the negative impact of the COVID-19 pandemic on populations' well-being. Much of this literature focuses on the direct and indirect consequences of the pandemic—through the social, emotional and economic consequences of lockdowns and health policies—on mental health (Brooks et al. [19], Holmes et al. [20], Blasco-Belled et al. [21], Cooke et al. [22], Rajkumar [23], Xiong et al. [24], Saladino et al. [25], Li and Wang [26], Cao et al. [27]). Salari et al. [28]'s meta-analysis of published studies on mental health in the general population reveal that five studies indicate a prevalence of stress (30%), 17 studies indicate a prevalence of anxiety (31.9%), and 14 studies indicate a prevalence of depression (33.7%) (Please note these rates refer to the first wave of COVID-19 pandemic). Kawohl and Nordt [29] estimated the impact of the expected rise in unemployment during COVID-19 on suicide rates. Using data on expected job losses from the International Labor Organization and a previously developed model about unemployment (from 2000 to 2011)

and suicide risk, the authors expect that suicides could increase by about 9570 units per year if the unemployment rate increases from 4.9% to 5.6% (the high scenario). In the low case scenario (unemployment rising to 5.1%), suicides could increase by 2135 units per year. Krendl and Perry [30] found that older adults reported higher depression and greater loneliness following the onset of the pandemic in a sample of American respondents. Sibley et al. [31] documented an increase in anxiety/depression following lockdown and warned about long-term challenges to mental health in New Zealand. Patrick et al. [32] reported marked changes in the mental health of parents and children in the United States. O'Connor and Peroni [33] documented a decline in mental health for nearly a third of residents in Luxembourg. The most important factors associated with the decline in mental health were worsening physical health, income, and a decline in job security.

There are, however, some examples of contrasting results. Sønderskov et al. [34], for instance, documented an *increase* in the psychological well-being of the Danish population from the first wave (March 31 –April 6, 2020) to the second one (April 22 –April 30, 2020), probably because symptoms of anxiety and depression decreased. Recchi et al. [35] reached a similar conclusion. Using panel data from France (administered at three points in time between the 1st of April and the 6th of May 2020), the team found that, in general, self-reported health and well-being improved during the lockdown with respect to the previous year. Although the result hides some heterogeneity within the population (blue-collar workers seem to have suffered more from the crisis), the authors explained their finding, arguing that individuals not affected by the virus judged their situation better than they normally would have. As the authors warn, their findings are based on data from the first six weeks of lockdown in France, and they consider the possibility that the pandemic will affect the population in the long run.

Most studies indicate that well-being decreased in correspondence with the pandemic; however, the mechanisms are much less agreed upon. Rossouw et al. [36] show that lockdown regulations hampered happiness—measured by GNH—in South Africa and argue that the determinants of happiness under lockdown were directly linked to the implemented regulations: lack of access to alcohol (and tobacco), increased social media usage, concerns over future employment, and more time spent at home. Similarly, Greyling et al. [37] found a negative effect of lockdown on GNH in South Africa, New Zealand and Australia. Unobserved factors can confound the findings, however. To address this issue, Foa et al. [38] distinguish the effect of lockdown from that of the pandemic by using weekly data issued from YouGov's Great Britain Mood Tracker Poll and Google Trends. They found that lockdowns were positively related to subjective well-being and that the main threat to mental health was the severity of the pandemic. The authors also suggested that lockdowns helped relieve the negative impact of the pandemic on well-being by relieving anxiety and stress. Another study, Kivi et al. [39], shows increasing well-being (life satisfaction and reduced loneliness) during the pandemic (among older Swedish residents compared to previous years (2015–2020), but only during the early stage, and also decreasing well-being for those more worried about the pandemic.

With only a few exceptions, previous studies mainly observed well-being at a given point in time. This is because well-being is commonly measured with surveys that take time to administer and elaborate before the data can be published. Only a few studies used Big Data to track the evolution of well-being during the pandemic. As the impact of the pandemic on societies changes over time, it is important to monitor the dynamics of well-being to fully understand their causes and economic, social, and political consequences.

The few available studies on well-being changes during the pandemic reached varied conclusions. Wang et al. [40] found no differences in the Chinese population's stress, anxiety, and depression levels when comparing two waves (January 31—February 2, to February 28—

March 1, 2020). Brülhart and Lalive [41] compared helpline calls to Switzerland's most popular free helpline during the pandemic to the previous year's records to infer the impact of COVID-19 on people's suffering. They concluded that the impact of COVID-19 was negligible as the number of calls has grown in line with the long-run trend. Using GNH data from New Zealand, Rossouw et al. [42] documented a decline in well-being that persisted over time. Sibley et al. [31] reported limited well-being changes during the pandemic's first stages, possibly due to increased community connectedness offsetting otherwise negative impacts. Brodeur et al. [17] analysed data from Google Trends collected between the 1st of January 2019 and the 10th of April 2020 in nine Western European countries and the American States. Although their main aim was to assess the causal link between lockdown and the search intensity for various proxies (of lack) of well-being (terms included boredom, contentment, divorce, impairment, irritability, loneliness, panic, sadness, sleep, stress, suicide, well-being, and worry), they found evidence of mean-reversion in several measures of well-being, but not in all. They concluded that the level of well-being at the beginning of lockdown could be a poor guide to its level later. Cheng and colleagues [43] reached a similar conclusion. The team applied difference-in-difference techniques to monthly data from a longitudinal survey administered on middle-aged and older adults in Singapore. They found large declines in overall life satisfaction and domain satisfaction during the outbreak: the magnitude of the effects is comparable to those of a major health shock or the loss of a beloved person. The figures also indicate that the impact of the COVID-19 pandemic on life satisfaction extended for multiple months but attenuated with time, as life satisfaction trended towards pre-pandemic levels later in the year.

There are various possible explanations for the contrasting results reported above. For instance, previous studies are generally based on data for just one country, often comparing the same country over time or comparing the pandemic to a "normal" year, i.e., 2019. Such analyses hold broad institutional characteristics constant but prevent researchers from comparing these characteristics. Different country contexts, infection rates, and governmental responses are other potential explanations for the heterogeneity of results. This motivates our cross-country investigation over time.

## 2.2 Use of Big Data to construct happiness indices

Authors from many social sciences have applied Natural Langue Processing to Big Data to address various issues (Eichstaedt et al. [44], Caldarelli et al. [45], Gayo-Avello [46], Bollen et al. [47], Asur and Huberman [48], O'Connor et al. [49]). For instance, tweets have been used to track the rate of influenza in the United Kingdom and the United States (Lampos and Cristianini [50], Culotta [51]). Paul and Dredze [52] found a positive association between public health data and the data issued from sentiment analysis of tweets.

Only a few studies use Big Data to calculate a happiness index. The first measure, The Hedonometer, was created by Dodds and Danforth [53] and their team in 2008. They use the Twitter Decahose Application Programming Interface (API) feed, which is a streaming API feed that continuously sends a sample of roughly 10 per cent of all tweets. This sample allows Dodds and the team to measure happiness levels per day continuously, thus resulting in a time series from 2008 to the present (Dodds et al. [54]). However, the Hedonometer cannot deal with the context in which words are used. Individual words are evaluated, not the overall sentiment of the tweet. For example, a phrase such as "I did not enjoy the holiday" will attract a score of 7.66 for 'enjoy' and 7.96 for 'holiday', thus reflecting an overwhelmingly positive sentiment when actually the sentiment is negative. Furthermore, the Hedonometer calculates a happiness index on a scale of 1 (sad) to 9 (happy), but it cannot detect the emotions

underpinning the words or the tweets. Thus, it cannot determine if the changes in happiness levels are due to more or less negative emotions such as fear or anger or positive emotions such as joy.

The second known measure was developed in 2012 by Ceron et al. [55]. They used an Integrated Sentiment Analysis (a human-supervised machine learning method) on Big Data extracted from Twitter for both Italy and Japan. They created a composite index of subjective and perceived well-being that captures various aspects and dimensions of individual and collective life (Iacus et al. [56]). Up until 2017, the researchers extracted and classified 240 million tweets over 24 quarters. They applied a new human-supervised sentiment analysis to analyse the sentiment and did not rely on lexicons or special semantic rules.

## 3. Data and methodology to construct time-series data from tweets

This section describes the dataset used, the novel methods to construct daily time-series data from tweets using Natural Language Processing, and lastly, the selected variables used in further analyses.

### 3.1 Data: Applying natural language processing to Big Data

In the analyses, we use a cross-country panel dataset with high-frequency daily data (see section 3.2). We analyse the time period from 1 January 2020 to 31 December 2020 (366 days since 2020 was a leap year) across ten countries, namely Australia, Belgium, France, Great Britain, Germany, Italy, Luxembourg, New Zealand, South Africa and Spain.

To derive our data, we use the Twitter API (v1) to extract and harvest all original tweets within a geographic bounding box that corresponds with the country in question. After translating all non-English tweets into English, using Microsoft Azure and Google Translate, we performed necessary pre-processing to clean up the punctuation and remove @, #, the letters "https", control characters, digits, and emojis. We removed emojis because, in further analyses, we use only ASCII characters. We then use Natural Language Processing (NLP) in the form of sentiment and emotion analysis to score each tweet's underlying sentiment and emotions. Stated briefly, sentiment analysis is an automated process to determine the feelings and attitudes of the author of a written text (or tweet) (Hailong et al. [57]). The time series were then based on the average daily coded values described below.

Applying NLP to Big Data sources has some advantages over traditional tools, as discussed in the Introduction. One of the advantages of Twitter data is that they are abundant, and users are heterogeneous. Twitter accounts include individuals, groups of individuals, organisations, and media outlets, thus providing the moods of a vast blend of users, which is not found in survey data. Another advantage is that they can provide timely and internationally comparable data on individuals' well-being, sentiments, and behaviours. The data also does not suffer from non-response bias (Callegaro and Yang [58]). Sentiment analysis applied to social media data and location history allows researchers to "listen" and observe what people deem important in their lives. Big data can be used by decision-makers, especially in situations requiring rapid decisions or in the presence of incomplete information. However, using Big Data implies a trade-off between timeliness and accuracy. Big data sources may face large measurement errors and limited representativeness. It is also difficult to establish their convergent validity as there are very few validated data sources to compare.

### 3.1.1 The dependent variable: The Gross National Happiness index

To measure happiness (the dependent variable), we make use of the GNH, which was launched in April 2019 as part of the *Gross National Happiness.today* project (Greyling et al. [15]). GNH

measures the evaluative mood of a country's citizens over time. As a measure of mood, the GNH captures the more volatile part of well-being, commonly referred to as happiness (Diener et al. [59]). (Please refer to Rossouw and Greyling [60] for country examples where the GNH accurately captures the mood of a nation). However, the evaluative qualification indicates tweets reflect individuals' conscious decisions; in other words, they evaluate what they want to say.

The GNH index is constructed using sentiment analysis. After completing the pre-processing of the tweets, each tweet was coded as having either a positive, neutral or negative sentiment primarily using the lexicons Sentiment140 and NRC (National Research Council of Canada Emotion Lexicon developed by Turney and Mohammad [61]). For a full discussion on using sentiment and emotion analyses to derive variables, see Greyling and Rossouw [62]. Then, if a tweet is positive, it is coded as 0, if neutral 2 and if negative 4. The number of positive and negative tweets are used in a sentiment-balance algorithm (a balance score) and scaled to derive a GNH happiness score per hour that ranges from 0 to 10, with higher values indicating higher happiness. To generate daily data, the mean GNH per day is calculated.

### 3.1.2 Emotion variables

Emotions are derived using emotion analysis operationalized using the NRC lexicon. It distinguishes between eight basic emotions: anger, fear, anticipation, trust, surprise, sadness, joy and disgust (the so-called Plutchik [63] wheel of emotions). NRC codes words with different values, ranging from 0 (low) to 10 (high), to express the intensity of an emotion or sentiment. See Greyling et al. [37] and Greyling and Rossouw [62] for previous examples using these eight emotions.

Table 1 provides three examples of how emotions are extracted. Each tweet is attributed a score according to the presence and intensity of one or more emotions. For example, the tweet "*Mask-wearing is really reducing in inner Auckland–I've been virtually the only one I've seen today. (Lack of) distancing pretty much the same. . . .#COVID19NZ* "; resulted in a score of 6 being assigned to the emotion called 'anger', a score of 2 to the emotion called 'anticipation', a score of 2 to the emotion called 'surprise' and a score of 4 to the emotion called 'disgust'. The daily score of a given emotion corresponds to the average of the scores that emotion received on a given day. As an example, Table 1's tweets generate a score of 3.3 for 'anger', i.e. (0 + 4 + 6)/3.

To develop our context-specific variables, we extracted three subsets of tweets based on keywords relating to the economic situation, institutions, and loneliness. Emotion analysis using the same process and NRC lexicon was then conducted on these subsets to generate the specific variables:

**Table 1. Examples of coding tweets for emotions.**

| *"I love dogs; they are such good companions"* | | | | | | | |
|---|---|---|---|---|---|---|---|
| Anger | Fear | Anticipation | Trust | Surprise | Sadness | Joy | Disgust |
| 0 | 0 | 0 | 1 | 0 | 0 | 2 | 0 |
| *"Judith's doing a great job boosting the party vote in her new role as leader of the Nat Party, hope they get rid of that Bridges guy now"* | | | | | | | |
| Anger | Fear | Anticipation | Trust | Surprise | Sadness | Joy | Disgust |
| 4 | 0 | 1 | 2 | 0 | 0 | 5 | 0 |
| *"Mask-wearing is really reducing in inner Auckland–I've been virtually the only one I've seen today. (Lack of) distancing pretty much the same. . . .#COVID19NZ"* | | | | | | | |
| Anger | Fear | Anticipation | Trust | Surprise | Sadness | Joy | Disgust |
| 6 | 0 | 2 | 0 | 2 | 0 | 0 | 4 |

Source: (Greyling et al. [15]).

1. To construct the 'fear in relation to the economic situation' variable, we first extract all tweets that include the following keywords: *jobs*, *economy*, *saving*, *work*, *wages*, *income*, *inflation*, *stock market*, *investment*, *unemployment*, *unemployed*, *employment rate*, *tech start-up*, *venture capital*. We then use the *NRC lexicon* to return the fear emotion score for each tweet in this subset.

2. To construct the 'sadness in relation to loneliness' variable, we first extract all tweets that include the following keywords: *lonely*, *loneliness*, *alone*, *isolation*, *abandoned*, *social distancing*, *lonesome*, *by oneself*, *solitary*, *outcast*, *companionless*, *solitary*, *homesick*. We use the same method to construct the time-series as for economic fear, but now we use the emotion score for 'sadness' and the loneliness subset.

3. To construct the 'trust in national institutions' variable, we first extract all tweets that include the following keywords: *parliament*, *ministry*, *minister*, *senator*, *MPs*, *legislator*, *political*, *politics*, *prime minister*. We use the same method as above, but this time we use the emotion score for 'trust' and the national institutions subset.

### 3.1.3 Validity of GNH and the emotion variables

It is vital to assess whether GNH reliably measures the aggregate happiness of a nation. The validity of the GNH can be assessed in several ways. We assume that content validity, i.e., the ability of the GNH to represent happiness correctly, is satisfied. The reason is that the algorithm for sentiment analysis is logically built to measure the affection content of a tweet. Sensitivity to variations in the algorithms used, volumes of tweets, and the sample period are also assessed, with results presented in S1 Appendix. We test the convergent validity of the GNH by checking whether GNH significantly correlates with external data that are known to represent the same or similar concepts.

To assess the external convergent validity, we proceed as follows: firstly, we compare cross-sectional country rankings produced from the GNH to those available from alternative measures of well-being, such as the Eurobarometer's life satisfaction and the World Happiness Report (Helliwell et al. [64]). Secondly, we analyse the time-series correlation between GNH changes and changes in consumer confidence and other well-being indicators available for the same period from Google Trends. This exercise provides encouraging results: GNH-based country rankings are not significantly different from those based on alternative indicators. Results from time-series correlations are mixed: the correlation of GNH to well-being from survey data for four European countries is uncertain. At the same time, it performs relatively well in relation to the index of negative emotions and consumer confidence. The validation results pertaining to external data are presented in S2 Appendix. Similar validity tests of generalised trust, trust in national institutions, and economic fear are contained in S5 Appendix. Indeed, they show significant relations that support the overall process by which the Twitter-derived variables are generated.

In addition to our tests, previous literature suggests that the GNH index correctly reflects the evaluative mood of a nation (i.e., supporting construct validity). Greyling et al. [37] showed a negative and statistically significant association between the GNH index and 'depression' and 'anxiety' in Australia, New Zealand and South Africa. Moreover, data indicate that variations in the GNH index correctly reflect various events, including the COVID-19 pandemic. Data from South Africa show that the GNH dropped well below previous daily averages following the outbreak of COVID-19. Later, when distancing regulations were implemented, the GNH recovered slightly but remained lower than normal (Greyling et al. [37]). In a different

**Table 2. Descriptive statistics by country.**  Average values over the year 2020.

| Country | GNH (0–10) | Confirmed cases (Cum.) per million | Containment policy (0–100) | Residential mobility (%) | Unemployment rate (%) | Consumer confidence Pos. Bal. (%) | Tweets per day |
|---|---|---|---|---|---|---|---|
| Australia | 7.27 | 1,115 | 54.62 | 8.58 | 7.4 | N/A | 24,354 |
| Belgium | 7.05 | 55,782 | 51.24 | 11.57 | 5.63 | -12.16 | 6,335 |
| France | 6.29 | 41,022 | 54.90 | 10.52 | 8.18 | -12.89 | 37,250 |
| Germany | 7.43 | 21,013 | 51.94 | 6.75 | 4.19 | -9.55 | 22,318 |
| Italy | 7.22 | 34,851 | 58.56 | 10.37 | 9.12 | -16.65 | 27,677 |
| Luxembourg | 7.14 | 74,148 | 43.16 | 12.29 | 6.75 | -11.69 | 257 |
| New Zealand | 7.06 | 448 | 35.48 | 8.00 | 4.55 | N/A | 4,624 |
| South Africa | 6.34 | 17,825 | 53.18 | 15.18 | 28.74 | N/A | 57,256 |
| Spain | 6.81 | 41,242 | 56.27 | 10.22 | 15.54 | -22.86 | 55,289 |
| United Kingdom | 7.42 | 36,771 | 57.06 | 13.16 | 4.20 | -16.56 | 93,500 |

Note: The unemployment rate is unadjusted, and consumer confidence is adjusted.

Source: all sources are described in the text. They are omitted for brevity.

field, Lampos and Cristianini [50] and Culotta [51] showed that information extracted from tweets could track health variables.

Additional concern may arise from assigning tweets to specific countries because that limits the number of usable tweets as not all tweets are geo-located. However, a large number of tweets are extracted per country, and the number of users represents significant proportions of the countries' populations. For example, the number of extracted tweets per day ranges from 257 in Luxembourg to 4,600 in New Zealand and as many as 93,500 tweets in the United Kingdom. The smaller number of tweets in Luxembourg is offset by the smaller population and the analytical techniques used to assess GNH. For instance, we generally focus on longer periods, e.g., weeks, or use rolling averages to smooth daily fluctuations. We also conducted several internal consistency tests (reported in S1 Appendix) to ensure the GNH is not too sensitive to changes in the volume of tweets.

In summary, previous studies and the validation exercises from this study provide initial evidence suggesting that the GNH provides a meaningful measure of happiness. Tests of three emotion variables also support the use of Twitter-derived variables. S5 Appendix gives details on the validity of these measures (Figs S17 and S18 in S1 Appendix). Descriptive statistics of the GNH are presented in Table 2.

## 3.2 Additional covariates

We integrate the variables derived from Twitter data with additional information to account for each country's evolution of the pandemic. These include:

1. Daily new cases: Data on COVID-19 are sourced from Our World in Data (Roser et al. [65]). Among available series, we retain the number of new confirmed cases per day per million in population. We adjust by population to account for countries' sizes. In much of the analyses, we further transform new cases using an inverse hyperbolic sine transformation, which is roughly equivalent to a log transformation but is identified for zeros. Please note that the European countries' data are from the ECDC (European Centre for Disease Prevention and Control). Available series also include the number of tests performed, deaths and hospitalisations.

2. Policy responses: The indicator of policy stringency is the Containment and Health Index from the University of Oxford's COVID-19 Government Response Tracker (Hale et al. [66]). The tracker includes multiple indices summarising 18 indicators of policy response to the COVID-19 pandemic in different dimensions. The Containment Index, also known as the stringency index, is based on the following nine indicators: school closing, workplace closing, cancel events, restrictions of gathering, close public transport, stay at home requirements, restrictions on internal movement, international travel controls, and public information campaigns. Details on the construction of the index and the underlying indicators are available online (www.bsg.ox.ac.uk/covidtracker).). The data and methodology are frequently updated as the pandemic evolves.

3. Behavioural responses (distancing): We use Google Mobility Reports (Google [67]) to measure distancing and account for behavioural responses to the pandemic and government policies. Google Mobility Reports provide daily aggregate mobility/visitation data by geographic location. The data, collected from users' devices that have opted-in to location history on their Google account, permit the observation of mobility trends across several types of locations: retail and recreation, groceries and pharmacies, parks, transport hubs, workplaces, and residential. The measure of distancing we consider is an index reflecting the time people spend at home. Our choice is motivated by requiring fewer assumptions about people's movements during the pandemic. The figures are compiled as relative movements (visits' numbers) compared to the number of visits during the baseline period, 3 January to 6 February. Mobility is also normalised for each day of the week. Seven baseline days are used, corresponding to the median values observed during the five-week baseline period. For this reason, we cannot compare daily movements. We instead use weekly average values or daily data smoothed using a seven-day centered moving average.

4. Economic conditions (unemployment rate): The monthly unemployment rate is made available for the European countries by Eurostat (Eurostat, [68]). We use the raw, not seasonally adjusted series.

Table 2 provides summary statistics for the non-Twitter variables described in points 1–4 above, as well as information on the GNH.

## 4. Methodology

Previous literature indicates that the pandemic negatively affects well-being through multiple channels. To simultaneously account for the role of the various possible explanatory factors related to changes in the GNH, we use multivariate ordinary least squares regressions. For policies, we distinguish between the effect of an expected increase in containment policies from the one of an expected decrease. We also assess the role of physical distancing, economic fear, trust in national institutions, loneliness, and generalized trust. We also include controls for the remaining emotions (anger, anticipation, disgust, fear, joy, sadness, and surprise, besides generalized trust) and seasons and months to account for unobserved factors such as weather.

The estimated regression model, which accounts for the time-series properties of the data, is given by the following equation:

$$
\begin{aligned}
GNH_{it} = {} & \alpha + \rho GNH_{it-1} + \beta_1 IHS(Cases)_{it} + \beta_2 Distancing_{it} + \beta_3 Decr.Cont.Policies_{it+1} \\
& + \beta_4 Incr.Cont.Policies_{it+1} + \beta_5 Emotions_{it} + \beta_6 EconFear_{it} + \beta_7 GenTrust_{it} \\
& + \beta_8 InstTrust_{it} + \beta_9 Loneliness_{it} + \beta_1 0 X_{it} + \epsilon_{it}
\end{aligned}
\tag{1}
$$

Where $GNH_{it}$ represents the average Gross National Happiness for country $i$ in week $t$. $IHS$ (*Cases*) represents the inverse hyperbolic sine of the average number of new cases per million

in a week. *Distancing* is the index of residential mobility. *Decr. Cont. Policies$_{it+1}$* is the expected decrease of the containment policy index in the following week. It is a dummy variable set to one if the index decreases at *t+1*, zero otherwise. In detail, *Decr. Cont. Policies$_{it+1}$* = 1 *if Cont. Policies$_{it+1}$−Cont. Policies$_{it}$*<0; *or Decr. Cont. Policies$_{it+1}$* = 0 *if Cont. Policies$_{it+1}$−Cont. Policies$_{it}$*≥0

*Incr. Cont. Policies$_{it+1}$* is the reciprocal: the expected increase of the policy index in the following week. It is set to one if containment policies increase at *t+1*, zero otherwise. We also control for economic fear, generalised trust, trust in national institutions and loneliness. *Emotions* is a vector including the emotions previously mentioned. *X* is a vector of control variables, including dummies for each month and season.

Statistical significance is assessed using Wild Cluster Bootstrap methods. Clustering standard errors at the country level are necessary because of the strong persistence in both the dependent and independent variables within a country. Bootstrap methods are needed because the number of countries is small. A small number of clusters leads to rejecting the null hypothesis relatively more frequently, in some cases at more than double the critical value (Bertrand et al. [69]). Wild Cluster Bootstrap methods resample over clusters and, using Webb weights, are particularly intended to accommodate scenarios with less than ten clusters. The limitation of the Wild Cluster Bootstrap method is that only the p-values from the bootstrap distribution can be obtained to assess the significance of coefficients (Cameron and Miller [70]).

## 5. Results

### 5.1 Descriptive analysis

Before turning to the regression analysis results, we describe our variables' changes during the year 2020. For brevity, we group countries into two groups: the seven European countries (EU hereafter), and Australia, New Zealand and South Africa (A-NZ-SA hereafter). Please note detailed variables' evolution for each country is presented in S3 Appendix.

Fig 1 depicts the time series of the GNH (solid line) for the EU countries (panel 1a) and A-NZ-SA (panel 1b) for 2020. Noticeably, in the EU, the GNH dipped at the onset of the first and second waves of the pandemic. The fall in the GNH is also apparent for A-NZ-SA at the beginning of the first wave. The figure also illustrates the dynamics of GNH throughout the

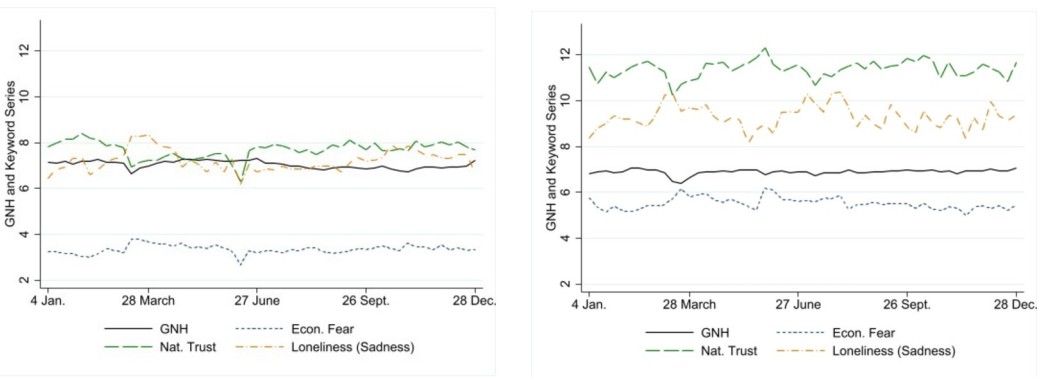

**Fig 1. Gross National Happiness, economic fear, trust in national institutions, and loneliness in 2020.** Note: Data are presented using seven-day (centered) moving averages. Figures for Luxembourg are missing due to the scarcity of tweets using the selected keywords. Source: Data are all sourced from the project "Preferences Through Twitter" with the support of FNR, UJ and AUT. a) Average daily data across six European countries. b) Average daily data across Australia, New Zealand and South Africa.

year and how measurements based on a single point in time could misrepresent the dynamics (e.g., mid-March or mid-May).

Fig 1 also shows the time series data for economic fear, loneliness and trust in national institutions. Please note that the number of European countries decreases to six because figures are unavailable for Luxembourg.

One can see from Fig 1 that A-NZ-SA experienced higher levels of economic fear, sadness linked to loneliness, and trust in national institutions than the EU. Trust and sadness exhibit higher volatility in A-NZ-SA than in the EU. We notice a marked decrease in trust in national institutions in correspondence with the first severe coronavirus outbreak (March), followed by a slow recovery in both groups of countries. At the same time, loneliness increased, with a further increase in EU countries during the second wave. Economic fear increased during the first wave. Afterwards, it remained at levels higher than the initial one in the EU countries. At the same time, it decreased slightly but steadily in A-NZ-SA (S15 Fig in S3 Appendix provides detailed trends for each variable for each country). The limited number of tweets produced in Luxembourg did not allow us to compute trust in national institutions, sadness in relation to loneliness, and economic fear.

Fig 2 contrasts the respective evolutions of average GNH (solid line) and COVID-19 infections (dashed line) for the EU (panel 2a) and A-NZ-SA (panel 2b). Please note that infections are the average number of daily new confirmed positive cases per million. Note also that mass testing was not performed during the first pandemic wave.

Panel 2(a) shows that GNH dipped in correspondence to the two pandemic peaks of March and November 2020. In Europe, GNH dropped suddenly (-8.6%) during the first wave, recovering quickly afterwards (+9.84%). In correspondence to the slow but steady increase in the number of cases during the late European summer-autumn, GNH showed a steady decline culminating with a sharp fall at the beginning of November, when infections reached a second peak.

Panel 2(b) shows a similar pattern for A-NZ-SA's GNH during the first peak (GNH suddenly dropped by 9.33%). The evolution of GNH declined slightly during the emergence of the second pandemic wave (May-July) and recovered afterwards. We also observe that the number of new positive cases has been substantially lower than those recorded in the EU in this group of countries. Changes in the GNH in 2020 are more volatile in the EU than in A-NZ-SA (S4 Table and S16 Fig in S4 Appendix provide average scores of the GNH by sub-periods, while

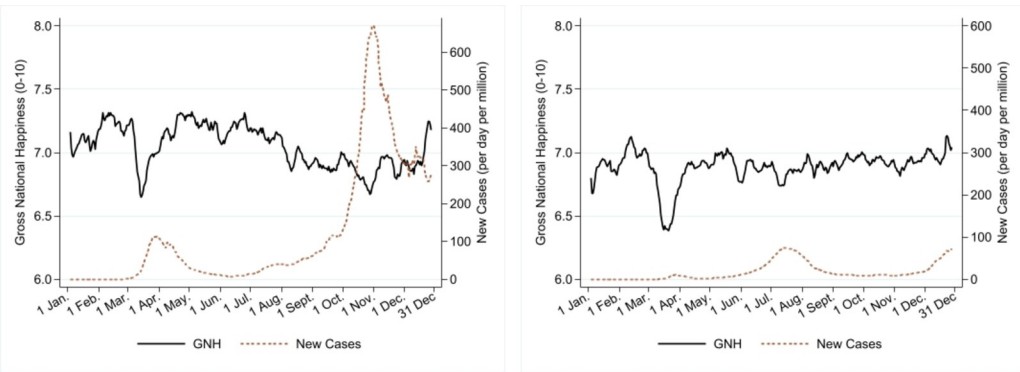

**Fig 2. Gross National Happiness and new COVID-19 cases per day in 2020.** Note: GNH and new cases are presented using seven-day (centered) moving averages. Source: GNH data (Greyling et al. [15]) are sourced from the project "Preferences Through Twitter" with the support of FNR, UJ and AUT. The number of new positive cases is sourced from OurWorldinData.org. A) Average daily data across seven European countries. B) Average daily data across Australia, New Zealand and South Africa.

the changes of GNH for each country are shown in S13 Fig in S3 Appendix). Hence, while the average GNH is about 7, our daily data reveal a varied picture in terms of intensity and duration of the shock and across waves of infection.

However, other variables may influence GNH besides the number of infections. Containment policies, economic conditions, trust in others and institutions may have affected the overall well-being, both directly and through their impact on the pandemic. We account for the joint effect of these variables using regression analysis in section 4.2. In the remainder of this section, we briefly describe changes in the GNH in relation to changes in containment policies and trust in others. These variables are relevant for their direct effects on well-being and because of their effect on the pandemic: containment policies limited the spread of COVID-19 (Fong et al. [71], Chinazzi et al. [72]), thus benefiting well-being. Trust in others promoted cooperation and solidarity with positive spillovers on compliance and well-being (Bargain and Aminjonov [73]).

Fig 3 reports the changes in the GNH (solid line) and those of containment policies (dashed line). We notice the jump in policy stringency and a drop in the GNH, which occurred at the pandemic's onset. After this, the months of May–July were characterised by a gradual relaxation of containment policies. Then, policies in the two groups of countries took different directions. In A-NZ-SA, increases in stringency during July-September were followed by a marked relaxation of policies. The summer coincided with a relaxation of policies in EU countries, down to a degree of stringency maintained throughout October. Stringency jumped again from about 50 to nearly 80 points during October, accompanied by the sharp fall in the GNH. However, the decline in the GNH had begun earlier. In A-NZ-SA, the increase in stringency saw a dip in GNH, but also, in this case, the latter's decline had started before.

It is worth noting that countries' policy responses to the pandemic have been widely heterogeneous, both within and between the two groups of countries. S14 Fig in S3 Appendix depicts changes in the GNH, the number of new positive cases and the level of containment policies separately for each country. The graphs show that policy response in the EU shares a similar pattern across countries–after the initial shock, stringency in containment policies increased with a rise in new positive cases. In contrast, countries in the Southern hemisphere saw nearly zero new infections, contrasted with heterogeneous governments' responses. Australia maintained strict containment policies throughout 2020, whereas South Africa gradually relaxed

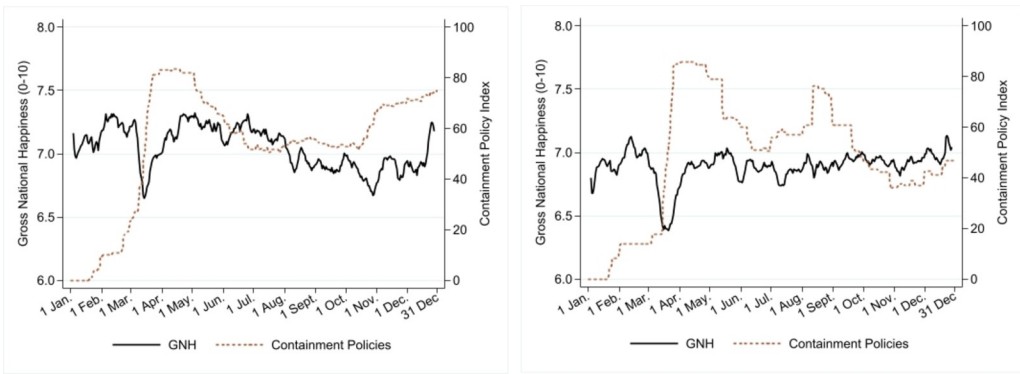

**Fig 3. Gross National Happiness and containment policies.** Average daily data across ten countries. Note: GNH is presented using seven-day (centered) moving averages. Source: GNH data (Greyling et al. [15]) are sourced from the project "Preferences Through Twitter" with the support of FNR, UJ and AUT. The policy index is sourced from Oxford Policy Tracker. a) Average daily data across seven European countries. b) Average daily data across Australia, New Zealand and South Africa.

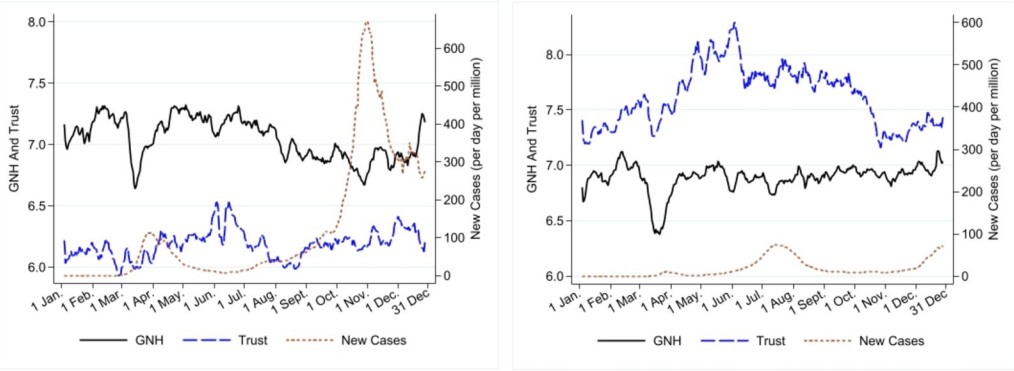

**Fig 4. Gross National Happiness, broad trust and new positive cases of COVID-19 in 2020.** Note: Data are presented using seven-day (centered) moving averages. Source: GNH data (Greyling et al. [15]) and generalised trust are sourced from the project "Preferences Through Twitter" with the support of FNR, UJ and AUT. The number of new positive cases of COVID-19 is sourced from OurWorldinData.org. a) Average daily data across seven European countries. b) Average daily data across Australia, New Zealand and South Africa.

the measures. In New Zealand, the initial suppression strategy was followed by a sharp drop in stringency, interrupted only by a sharp increase in stringency during August-September.

The levels of and changes in trust differ in the two groups of countries (see Fig 4). A-NZ-SA report, on average, higher trust in others than EU countries (respectively, 7.6 and 6.2 throughout 2020). Trust increased in both groups of countries in the first half of the year, markedly in A-NZ-SA (on average, 8.33% in European countries and 13.7% in A-NZ-SA). The upward trend was shortly interrupted in correspondence with the first outbreak of COVID-19. Subsequently, trust started declining (with an initial sudden drop of about 6%), corresponding to a renewed growth in infections (from June onward), with different patterns. In A-NZ-SA, it levelled off before decreasing again in September-October and started increasing again in November-December. In the EU, trust declined (nearly 7.7%) from June to September and exhibited an increasing trend afterwards. S15 Fig in S3 Appendix shows changes in trust for each country in the study. Generalised trust correlates at 81% (statistically significant at 10%) with survey-based measures of trust. See S19 Fig in S5 Appendix and the notes therein for more details.

In summary, the daily evolution of GNH reveals considerable variations in well-being responses during 2020. We found a clear indication that people suffer when the infection worsens and that the recovery takes longer than the decline. A descriptive examination of the changes in the number of new cases, containment policies, and generalised trust indicates a mixed relationship with changes in the GNH. This suggests that multiple factors should be considered jointly to explain the changes in well-being during the pandemic.

The next section explores the joint effects of multiple variables on the GNH using regression analysis.

### 5.2 Regression results

Following the methodology, as explained in section 4, we present the first set of results in Table 3. Regressors are included step-wise to check their association with GNH before and after controlling for additional variables. Coefficients on dummies for the months of the year are omitted for brevity.

The auto-regressive term, the lagged value of GNH, reported in the first row, has a high and significant coefficient, indicating that the current GNH variation depends largely on its

**Table 3. Association between the number of positive cases, physical distancing, expected increase and decrease of policy stringency, and GNH.**

| Variable | (1) | (2) | (3) | (4) | (5) | (6) |
|---|---|---|---|---|---|---|
| Lag GNH | 0.907 | 0.906 | 0.93 | 0.908 | 0.922 | 0.906 |
| | [0.000] | [0.000] | [0.000] | [0.000] | [0.000] | [0.000] |
| Δ IHS New Cases | | -0.064 | | | -0.039 | -0.048 |
| | | [0.010] | | | [0.007] | [0.021] |
| Residential—mobility | | | 0.006 | | 0.004 | |
| | | | [0.000] | | [0.003] | |
| F. Decr. Stringency | | | | -0.026 | -0.032 | -0.029 |
| | | | | [0.079] | [0.022] | [0.043] |
| F. Incr. Stringency | | | | -0.051 | -0.049 | -0.044 |
| | | | | [0.005] | [0.007] | [0.035] |
| Spring | 0.197 | 0.173 | 0.186 | 0.199 | 0.175 | 0.18 |
| | [0.078] | [0.110] | [0.058] | [0.082] | [0.090] | [0.107] |
| Summer | -0.013 | -0.012 | 0.005 | -0.004 | 0.01 | -0.004 |
| | [0.339] | [0.270] | [0.613] | [0.679] | [0.696] | [0.689] |
| Fall | 0.179 | 0.157 | 0.174 | 0.189 | 0.17 | 0.17 |
| | [0.087] | [0.115] | [0.045] | [0.091] | [0.093] | [0.117] |
| Constant | 0.63 | 0.643 | 0.486 | 0.652 | 0.536 | 0.664 |
| Month Controls | yes | yes | yes | yes | yes | yes |
| N | 510 | 510 | 460 | 500 | 450 | 500 |
| Adj. R Sq. | 0.837 | 0.839 | 0.842 | 0.844 | 0.849 | 0.845 |
| # of Countries | 10 | 10 | 10 | 10 | 10 | 10 |

Bootstrapped p-values in brackets.

previous realisations. Lagged GNH, along with the month and season controls, explains nearly 84% of the overall variance. This indicates that the GNH is highly persistent and not greatly affected by volatile events. Auto-correlation is common in time series. In this case, it also represents a desirable feature in a measure of well-being: "in a world of bread and circuses" (Deaton [74]), the GNH seems to capture the bread more than the circuses.

Dummies for Spring and Fall have positive and significant (10 per cent) coefficients, indicating that the GNH tends to be, on average higher during those seasons. The coefficient for Summer is not statistically different from zero. Models 2 to 4 add, respectively, new cases, the index of residential mobility, and the expected changes in containment policy to the baseline (model 1). We find a negative and significant coefficient for new cases and both expected increase and decrease of containment policies. The index of residential mobility attracts a positive and statistically significant coefficient, indicating that, ceteris paribus, staying at home is positively related to GNH. Notice that the number of observations reduces by 50 when we control for mobility. The reason is that Google Mobility data are available from the beginning of the pandemic (mid-February); thus, the initial five weeks of 2020 are missing. Model 6 shows that the results of the complete model (column 5) do not depend on the smaller sample size due to the inclusion of the control for residential mobility.

Results do not change when we simultaneously include all of the controls (Model 5). The adjusted R-squared indicates that the full model explains 85% of the total variance, slightly improving the baseline. All coefficients maintain their sign, magnitude and statistical significance.

Table 4 presents results for our extended model, which includes the emotions, controls for economic fear, trust in national institutions and loneliness. For ease of comparison, the first

**Table 4. Regressions of GNH on the complete set of control variables.** Average weekly values by country.

| Variable | (1) | (2) | (3) | (4) | (5) | (6) | (7) | (8) | (9) |
|---|---|---|---|---|---|---|---|---|---|
| Lag GNH | 0.922 | 0.756 | 0.76 | | 0.788 | 0.788 | 0.778 | 0.788 | 0.776 |
| | [0.000] | [0.002] | [0.000] | | [0.001] | [0.001] | [0.001] | [0.001] | [0.001] |
| Δ IHS New Cases | -0.039 | -0.03 | -0.027 | -0.067 | -0.031 | -0.03 | -0.031 | -0.031 | -0.032 |
| | [0.007] | [0.005] | [0.025] | [0.092] | [0.011] | [0.021] | [0.014] | [0.010] | [0.020] |
| Residential—mobility | 0.004 | 0.007 | 0.007 | 0.001 | 0.006 | 0.006 | 0.006 | 0.006 | 0.006 |
| | [0.003] | [0.001] | [0.000] | [0.975] | [0.000] | [0.000] | [0.001] | [0.000] | [0.000] |
| F. Decr. Stringency | -0.032 | -0.027 | -0.026 | -0.028 | -0.017 | -0.017 | -0.017 | -0.017 | -0.016 |
| | [0.022] | [0.166] | [0.124] | [0.975] | [0.184] | [0.218] | [0.207] | [0.118] | [0.145] |
| F. Incr. Stringency | -0.049 | -0.036 | -0.034 | -0.032 | -0.03 | -0.03 | -0.031 | -0.03 | -0.031 |
| | [0.007] | [0.018] | [0.012] | [0.348] | [0.013] | [0.001] | [0.016] | [0.014] | [0.004] |
| Anger | | -0.079 | | | | | | | |
| | | [0.657] | | | | | | | |
| Anticipation | | -0.075 | | | | | | | |
| | | [0.319] | | | | | | | |
| Disgust | | -0.13 | -0.28 | -1.376 | -0.357 | -0.344 | -0.332 | -0.359 | -0.337 |
| | | [0.519] | [0.000] | [0.106] | [0.003] | [0.014] | [0.016] | [0.003] | [0.019] |
| Fear | | -0.051 | -0.124 | -0.025 | -0.041 | -0.037 | -0.056 | -0.042 | -0.064 |
| | | [0.220] | [0.219] | [0.918] | [0.407] | [0.533] | [0.170] | [0.433] | [0.178] |
| Joy | | 0.048 | | | | | | | |
| | | [0.366] | | | | | | | |
| Sadness | | -0.12 | | | | | | | |
| | | [0.642] | | | | | | | |
| Surprise | | 0.327 | 0.329 | 0.942 | 0.301 | 0.3 | 0.289 | 0.302 | 0.286 |
| | | [0.040] | [0.037] | [0.027] | [0.038] | [0.533] | [0.170] | [0.433] | [0.026] |
| Trust | | 0.128 | 0.11 | 0.343 | 0.102 | 0.107 | 0.127 | 0.102 | 0.13 |
| | | [0.124] | [0.103] | [0.423] | [0.036] | [0.028] | [0.029] | [0.035] | [0.010] |
| Economic Fear | | | | | | -0.009 | | | 0.009 |
| | | | | | | [0.854] | | | [0.813] |
| Nat. Trust | | | | | | | -0.011 | | -0.014 |
| | | | | | | | [0.786] | | [0.595] |
| Loneliness (Sad) | | | | | | | | 0.001 | 0 |
| | | | | | | | | [0.933] | [0.993] |
| Constant | 0.536 | 1.074 | 0.966 | 5.127 | 0.801 | 0.77 | 0.84 | 0.803 | 0.882 |
| Month Controls | yes | yes | yes | - | yes | yes | yes | yes | yes |
| Season Controls | yes | yes | yes | - | yes | yes | yes | yes | yes |
| N | 450 | 450 | 450 | 450 | 405 | 405 | 405 | 405 | 405 |
| Adj. R Sq. | 0.849 | 0.881 | 0.88 | 0.527 | 0.928 | 0.928 | 0.928 | 0.927 | 0.927 |
| # of Countries | 10 | 10 | 10 | 10 | 9 | 9 | 9 | 9 | 9 |

Bootstrapped p-values in brackets.

column reports the same specification as column 5 in Table 3. Also, in this case, we included variables step-wise. We applied a step-wise selection process for the emotions to minimise the number of controls and preserve degrees of freedom. We kept the emotions with a p-value (after Wild Cluster Bootstrap) smaller than 0.4. After this selection process, only disgust, fear, surprise and trust are retained (column 3). Months and season controls are included in the estimates but omitted from the table.

The full model's adjusted R squared indicates that 92.7% of the total variance is explained, an improvement over the initial model of nearly seven percentage points, despite the decrease in the number of countries. Luxembourg is excluded from the analysis because tweets about economic conditions, national institutions, and loneliness are scarce.

The auto-correlation term still explains a large part of this variability, but its coefficient decreased from 0.922 to 0.776. The changes in the number of new infections and the expected increase of containment policies maintain their negative sign, magnitude and significance. The expected decrease in the policy stringency index is no longer statistically significant. Ceteris paribus, an increase in the number of people staying at home, remains positively and significantly associated with GNH changes.

GNH also increases when trust and surprise increase and decreases when disgust increases. These results are not surprising: it is well established that trust correlates with well-being both cross-sectionally and over time. Surprise and disgust are emotions that change with mood and correlate with more volatile aspects of well-being, sometimes referred to as affective or momentary subjective well-being. Economic fear, trust in national institutions, and sadness about loneliness are not significant. The reasons for this result are not clear.

In a further specification, we split changes in new positive cases into two variables: increases and decreases in new cases. This allows us to study the symmetry of the effect of contagion on the GNH (results are omitted for reasons of space and can be made available upon request to the authors). We found that an increase in new cases correlates negatively and significantly with the GNH, whereas a decrease does not attract a statistically significant coefficient. This suggests that a worsening pandemic affects the GNH much more than an improvement.

If we compare model 3, which includes Luxembourg, with model 5, we observe that the exclusion of Luxembourg increases our ability to explain the overall variance by 4.8 percentage points (from 88% to 92.8%). This is probably because the time series for Luxembourg are more volatile than those of the other countries. The exclusion of Luxembourg does not qualitatively change the estimated relations.

Model 4 is identical to model 3, except that it excludes the autoregressive term and month and seasonal controls. It is intended to check the robustness of results when increasing the degrees of freedom and removing the strong influence of lagged GNH. As expected, the adjusted R squared decreases considerably, from 88% to 52.7%. The coefficients of the variables of interest maintain their signs, but most of them lose significance: only the change in the number of new positive cases and surprise remain statistically significant.

In sum, our final model seems to explain the GNH changes during 2020 in the studied countries rather well. A large part of the overall variation is explained by the autoregressive term—which is to be expected and signals that the GNH is rather stable throughout the weeks of the year—as well as the month and seasonal controls. GNH decreases when the virus spreads, particularly when new positive cases increase and containment measures become more stringent. Under these circumstances, an increase in people staying at home is associated with an increase in the GNH. This is probably because people feel safe if they stay at home in a difficult or dangerous situation. Disgust and the GNH are negatively associated, whereas we find a positive association with surprise and trust.

## 6. Discussion

The following discusses the limitations of the data and generalizability and the results' implications.

Social media data offers advantages over traditional sources of statistical information; however, their validity and representativeness may limit how much we can generalize from the

results. Social media data only reflect the behaviours and characteristics of users, not the population as a whole. One concern relates to self-selection into the technology or platform (e.g., users of social networks may be systematically younger, better educated, or introverted than others). A further concern is that people's favourite social networks may change over time (Twitter could be widely used at time t and abandoned at time t+1). Another potential limit of Big Data is "fashion", or habit. Suppose users switch among social media platforms over time. In that case, the collected data may provide a good picture of the population at time t, but not at time t+1. These potential issues cast doubt on the generalizability and representativeness of the results from social media.

The degree of internet penetration and the "popularity" of Twitter in the countries studied should mitigate these concerns. In the countries we considered, internet penetration is very high (above 90%)—descriptive statistics about Twitter use are reported in Table 2. Each reassures us that, in principle, anybody could adopt Twitter. This, however, does not solve the problem of self-selection and representativeness. To the best of our knowledge, we cannot address these issues directly. We recommend that the validity of data sourced from social media should be carefully assessed and evaluated at every exercise. For instance, data from Twitter could be contrasted with alternative data, such as those issued from surveys, other Big Data or social media platforms, such as Google Trends. As discussed earlier, however, the possibilities to make such comparisons are limited, mainly because third-party high-frequency data are scarce.

To improve data validity and testing, researchers could use additional sources of social media data and, when using Twitter, further assess information from users' geo-locations and profiles. Using data from other social networks could mitigate the risk of self-selection into specific platforms, but unfortunately, we could not collect data from other social networks, such as Facebook and Instagram. Twitter offers an accessible API and privacy policies that allow researchers to analyze tweets for research purposes. It is worth noticing, however, that even if we could access all social networks available, some self-selection remains. It is possible that certain segments of the overall population make little or no use of social networks. Future research could also include self-reported locations provided by Twitter users in their profiles or posts to increase geographic precision. Another opportunity for future research is to sort individual (private) users from institutions and companies. Currently, however, most Twitter users are individuals, so the presence of tweets from other sources should have a limited impact on the final GNH scores.

The derivation of emotions in certain contexts could also be further developed. For instance, the feeling of sadness in a set of tweets that contained loneliness keywords is higher than in the full set of tweets. By construction, this context variable allows us to capture the amount, or degree, of sadness associated with loneliness and how this amount changed over time. Another potential variable, or alternative definition, could include the frequency of tweets about loneliness, which is also likely to have changed over time. While we believe that the novelty of our approach is valuable, there are certainly ways in which it can be expanded in future research.

Despite these limits, we believe the variables we constructed are new and have relevant implications for both policy and the social sciences. Our work makes two contributions from the research standpoint. First, it shows that collecting information about trust in institutions, economic fear, and sadness in relation to loneliness is feasible by applying sentiment analysis to tweets selected using specific keywords. Second, it provides a finer understanding of how the pandemic affected happiness and through which channels. The study provides information on the trajectory of happiness changes, the shape of the impact, and the speed and size of the subsequent recovery. From a policy standpoint, our work offers a method to satisfy the need for timely and relevant information.

Moreover, in times when survey answer rates are dropping, our work contributes knowledge on alternative data sources to monitor people's preferences and attitudes in nearly real-time. The pandemic was a "stress test" on the resilience of statistical offices: prolonged periods of lock-down disrupted the ability to collect information on what was happening in society. Our work shows that sentiment analysis of Big Data can provide useful information when alternative data sources are not viable. Lastly, we believe that our findings provide policymakers with information about the channels through which the pandemic can affect people's well-being. For instance, new positive cases and the stringency of containment measures negatively affected subjective well-being. This information can help policymakers respond to future health crises.

## 7. Conclusions

This was the first study to describe and explain changes in happiness throughout 2020—the year marked by the outbreak of the novel coronavirus pandemic—using high-frequency daily data in countries across Europe and the Southern hemispheres, specifically: Australia, Belgium, France, Germany, Great Britain, Italy, Luxembourg, New Zealand, South Africa and Spain.

For this purpose, we created a unique dataset using tweets extracted from Twitter. We apply Natural Language processing to score the tweets' sentiment and emotions. Using these scores, we derive high-frequency daily data in almost real-time. In this paper, we also use a novel method of filtering tweets on specific keywords, such as loneliness, and we derive the emotions of these tweets. Using the emotion scores on the keyword tweets, we derived the following variables per day: economic fear, trust in national institutions, and sadness in relation to loneliness. To the authors' knowledge, this is the first study to determine the aforementioned three variables using emotion analysis and subsequently use them in an empirical study. Additionally, no other study has performed the same extensive set of validation exercises on variables generated using Twitter data and sentiment and emotion analysis. In this way, this study contributes to the literature using Big Data and machine learning to compile and analyse social and economic data. The generated data can also valuably complement survey data to provide insights for the general public, the research community, and policymakers. This general interest to policymakers and statistical offices is applicable beyond the current pandemic outbreak.

The validation exercises provide initial evidence that the GNH provides meaningful information on national happiness. It generally correlates in the expected direction (though not always significantly) with alternative measures of well-being, and ill-being, from surveys and other Big Data sources, such as Google Trends. The same holds for economic fear, trust in national institutions, and generalised trust.

Our descriptive analysis showed that well-being exhibited considerable variation over the studied year. The first pandemic wave featured a sudden decline in the GNH, followed by a rapid recovery in all countries. Following this, the evolution of the GNH exhibited varied patterns across countries. In particular, the second wave of contagion was accompanied by a prolonged decline in the GNH in Europe. In Australia, New Zealand, and South Africa, a second period of decline in the GNH started in mid-May. It reached a peak at the beginning of July before recovering to its pre-pandemic levels.

To account for the simultaneous effect of various factors on the changes in GNH over time, we used regression analysis. Once accounting for the time series structure of the data and seasonal factors, we found that changes in the GNH correlate negatively with changes in new positive cases (and, in particular, the increases) and with the expected increase in containment policy stringency. An increase in people staying at home also correlates with an increase in the

GNH. This can be explained by an increased sense of protection and "altruism"—intended as own contribution to the fight against the spread of the virus—associated with increased distancing. Results also indicate that economic fear, trust in national institutions and sadness related to loneliness are not significantly associated with changes in the GNH. This is puzzling but could indicate that health and lockdown concerns, as captured by new cases and policy stringency variables, dominated individuals' mood during the pandemic. Finally, we found that GNH correlates positively when surprise and generalised trust increase and disgust decrease. Among these variables, trust is particularly relevant, as previous studies showed that higher trust correlates with higher compliance with containment policies and contributes to social cohesion and economic activity.

In summary, changes in the GNH during the pandemic correlate significantly with new infections, policy stringency, residential mobility, and trust. These correlations suggest that the GNH covers both cognitive and affective aspects of life as a measure of happiness.

## Supporting information

**S1 Appendix. Internal validity of the Gross National Happiness index.**
(DOCX)

**S2 Appendix. External validity of the Gross National Happiness index.**
(DOCX)

**S3 Appendix. Evolution of GNH and other variables by country.**
(DOCX)

**S4 Appendix. GNH evolution in different periods.**
(DOCX)

**S5 Appendix. Validity of additional variables obtained using Twitter.**
(DOCX)

**S1 Data.**
(DTA)

## Acknowledgments

We would like to thank AFSTEREO for the I.T. service provided.

## Author Contributions

**Conceptualization:** Francesco Sarracino, Talita Greyling, Kelsey O'Connor, Chiara Peroni, Stephanié Rossouw.

**Data curation:** Talita Greyling, Stephanié Rossouw.

**Formal analysis:** Francesco Sarracino, Kelsey O'Connor, Chiara Peroni.

**Funding acquisition:** Francesco Sarracino, Talita Greyling, Stephanié Rossouw.

**Investigation:** Francesco Sarracino, Kelsey O'Connor, Chiara Peroni.

**Methodology:** Francesco Sarracino, Kelsey O'Connor, Chiara Peroni.

**Project administration:** Francesco Sarracino, Talita Greyling, Stephanié Rossouw.

**Resources:** Francesco Sarracino, Talita Greyling, Chiara Peroni, Stephanié Rossouw.

**Software:** Francesco Sarracino, Kelsey O'Connor, Chiara Peroni.

**Supervision:** Francesco Sarracino.

**Validation:** Francesco Sarracino, Kelsey O'Connor, Chiara Peroni.

**Visualization:** Francesco Sarracino, Kelsey O'Connor, Chiara Peroni.

**Writing – original draft:** Francesco Sarracino, Kelsey O'Connor, Chiara Peroni.

**Writing – review & editing:** Talita Greyling, Stephanié Rossouw.

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
