## [Decision Letter · Decision Letter 0]

10 Mar 2022

PONE-D-21-37698

A year of pandemic: levels, changes and validity of well-being data from Twitter. Evidence from ten countries.

PLOS ONE

Dear Dr. Rossouw,

Thank you for submitting your manuscript to PLOS ONE. After careful consideration, we have decided that your manuscript does not meet our criteria for publication and must therefore be rejected.

Specifically:

1. Paper has failed to explain what the authors tried to communicate

2. Statistical analysis and results are insufficient

3. Contribution is insignificant

I am sorry that we cannot be more positive on this occasion, but hope that you appreciate the reasons for this decision.

Yours sincerely,

Debotosh Bhattacharjee, PhD

Academic Editor

PLOS ONE

Reviewers' comments:

Reviewer's Responses to Questions

**Comments to the Author**

1. Is the manuscript technically sound, and do the data support the conclusions?

Reviewer #1: Yes

Reviewer #2: Partly

2. Has the statistical analysis been performed appropriately and rigorously? 

Reviewer #1: Yes

Reviewer #2: No

3. Have the authors made all data underlying the findings in their manuscript fully available?

Reviewer #1: Yes

Reviewer #2: No

4. Is the manuscript presented in an intelligible fashion and written in standard English?

Reviewer #1: Yes

Reviewer #2: No

5. Review Comments to the Author

Reviewer #1: I would hereby recommend this paper for acceptance with minor revision because:

1. This work aimed to describe and explain changes in well-being that occurred in 2020 - using novel, timely data on happiness.

2. I feel that statistical analysis made by the authors are perfectly represented. In fact this paper represents perfectly the statistical measure made

3. Data representation is also done perfectly

4.I would hereby recommend that the authors can use multiclass SVM / some deep learning systems to represent their results. This in addition could be a more appropriate contribution.

Reviewer #2: (1) Presentation is poor.

(2) The paper is poorly formatted.

(3) From the sentence "This article analyses the well-being changes during 2020 using the Gross National Happiness.today

77 (GNH hereafter) index. The GNH is an aggregate country-level index of well-being, comparable across

78 countries (Greyling et al. [15]), compiled by applying sentiment analysis to Twitter posts (tweets)." ,

I can understand that sentiment analysis is applied in somewhere in the process. But it is unclear how the input to the sentiment analyzer is represented and how its output is fed to the overall decision making system.

(4) In a part of the paper, regression analysis has been mentioned without establishing any link to the earlier parts of the paper.

(5) In some parts of the paper, it seems that sentiment analysis was done manually, and in some other parts, it seems that it might be automatic. No diagram for the architecture of the overall system is given in the paper for resolving such confusion.

(6) From the sentence "The dataset also includes the following variables derived from the emotions expressed in tweets", it is not clear whether the values of the variables are manually derived or not. If it is manual, how can it be feasible for a large volume of social media data?

(7) It is expected that the role of sentiment analysis should be made clear in the motivation section also.

6. PLOS authors have the option to publish the peer review history of their article (what does this mean?). If published, this will include your full peer review and any attached files.

Reviewer #1: No

Reviewer #2: No

- - - - -

---

## [Author Response · Author response to Decision Letter 0]

3 Apr 2022

RESPONSE TO REVIEWERS

Manuscript ID PONE-D-21-37698

A year of pandemic: levels, changes and validity of well-being data from Twitter. Evidence from ten countries.

We thank the reviewers for the comments. Each has been comprehensively addressed, as set out below (authors' response shown in italics). Throughout the paper, we have also made improvements to enhance its quality and contribution to the literature. Therefore, we believe the paper to be significantly improved as a result of the comments received from the reviewers.

Reviewer comments addressed by the authors 

Reviewer #1: 

I would hereby recommend this paper for acceptance with minor revision because:

This work aimed to describe and explain changes in well-being that occurred in 2020 - using novel, timely data on happiness.

1. I feel that statistical analysis made by the authors are perfectly represented. In fact this paper represents perfectly the statistical measure made

We thank the reviewer for this generous comment.

2. Data representation is also done perfectly

We thank the reviewer for this generous comment.

3. I would hereby recommend that the authors can use multiclass SVM / some deep learning systems to represent their results. This in addition could be a more appropriate contribution.

We thank the reviewer for this suggestion. However, given the timeliness of the publication, we will take this comment onboard in future studies.

Reviewer #2: 

1. Presentation is poor.

We thank the reviewer for this comment and apologise for the poor presentation. Subsequently, we have made changes throughout the paper and significantly improved the presentation.

2. The paper is poorly formatted.

We thank the reviewer for this comment and apologise for the poor formatting. Subsequently, we have made changes throughout the paper and believe the format has improved. Please note that we have separated the results and methodology used, and each is located in its own section.

3. From the sentence "This article analyses the well-being changes during 2020 using the Gross National Happiness.today 77 (GNH hereafter) index. The GNH is an aggregate country-level index of well-being, comparable across 78 countries (Greyling et al. [15]), compiled by applying sentiment analysis to Twitter posts (tweets)." , I can understand that sentiment analysis is applied in somewhere in the process. But it is unclear how the input to the sentiment analyzer is represented and how its output is fed to the overall decision making system.

We thank the reviewer for this comment. However, we wish to clarify what sentiment analysis entails and how we measure happiness using the GNH index. 

We measure happiness by using the Gross National Happiness (GNH) index. The GNH measures the evaluative mood of individuals in our ten countries. We extract all geo-tagged tweets from Twitter and apply sentiment analysis to every tweet to construct the GNH index. 

Sentiment analysis is the process of determining whether a piece of writing conveys a positive, negative or neutral 'opinion'. We use Natural Language Processing (NPL) and a sentiment algorithm to determine the sentiment of a whole sentence rather than only the sentiment of a single word. This is the better choice as it helps you understand an entire opinion and not merely a word from the text. To determine the sentiment (and therefore the algorithm), we use two lexicons, namely Sentiment140 and NRC. 

The outcome of the sentiment analysis is used in an algorithm to derive the GNH per hour. The scale of the happiness scores is from 0 and 10, with 5 being neutral, thus neither happy nor unhappy. The index is available live on the GNH.Today website https://gnh.today and is also a formal statistic of life satisfaction reported by Statistics New Zealand; https://www.stats.govt.nz/experimental/covid-19-data-portal

Therefore, there is no input to a sentiment 'analyzer' because we use sentiment algorithms applied by our two lexicons, Sentiment140 and NRC. Additionally, there is no 'output fed to the overall decision making system' since we apply a sentiment balance algorithm to derive the hourly score, ranging from 0 to 10. The daily GNH score is the daily average. 

We note that there was no standalone methodology or data section that would have clarified the above from the onset. This has been rectified, and as a consequence, we believe our contributions have been enhanced further. We trust these are sufficient.

4. In a part of the paper, regression analysis has been mentioned without establishing any link to the earlier parts of the paper.

We thank the reviewer for this comment. As mentioned in point #3, we now have a distinguishable methodology section that addresses this comment. 

5. In some parts of the paper, it seems that sentiment analysis was done manually, and in some other parts, it seems that it might be automatic. No diagram for the architecture of the overall system is given in the paper for resolving such confusion.

We thank the reviewer for this comment. As stated throughout the paper, we use novel Big Data in our analyses. In layman's terms, Big Data is a phrase used to describe a massive volume of both structured (for example, stock information) and unstructured data (for example, social media postings) generated through information and communication technologies (ITC) such as the Internet. The volume of Big Data is so large that it is difficult to process using traditional database and software techniques. It is also unique in the velocity at which it is created and collected and the variety of the data points being covered. 

Given that we extracted from 4,600 tweets in New Zealand to as many as 93,500 tweets in the United Kingdom daily for the year 2020, it is impossible to complete sentiment analysis manually. The entire process of sentiment analysis is conducted using machine learning. 

Therefore, it implies that it is an automated process using different lexicons. Additionally, we wish to point out to the reviewer that we stated in the paper that sentiment analysis is an automated process (previously line 270, now line 275).

Lastly, we believe we have clarified what sentiment analysis entails and how it is conducted because of the change in structure.

6. From the sentence "The dataset also includes the following variables derived from the emotions expressed in tweets", it is not clear whether the values of the variables are manually derived or not. If it is manual, how can it be feasible for a large volume of social media data?

We thank the reviewer for this comment. As stated in points # 3 and #5, the values of the variables related to happiness and indeed daily observations (for a year) on emotions, trust in national institutions, loneliness and economic fear were all derived using machine learning. We use the NRC lexicon to determine the underlying tweets' emotions in terms of the emotion analyses. It distinguishes between eight basic emotions: anger, fear, anticipation, trust, surprise, sadness, joy and disgust (the so-called Plutchik wheel of emotions). NRC codes words with different values, ranging from 0 (low) to 10 (high), to express the intensity of an emotion or sentiment. 

We extracted our text corpus using specific keywords and applied emotion analyses to these tweets to determine the underlying emotion of economic fear, loneliness, and trust in national institutions. For example, to derive the economic fear variable, we first extracted all tweets that included the following keywords; jobs, economy, saving, work, wages, income, inflation, stock market, investment, unemployment, unemployed, employment rate, tech start-up, venture capital. Once we have our text corpus, we apply NRC and extract the emotion fear.

We included the above in Appendix D. However, due to changing the structure, we have added the above information in section 3.2.2. We believe it is now clearer to the reader.

7. It is expected that the role of sentiment analysis should be made clear in the motivation section also.

We thank the reviewer for this comment. As a consequence of addressing points #1 to #6, we believe this has been addressed.

---

## [Decision Letter · Decision Letter 1]

5 Jul 2022

PONE-D-21-37698R1A year of pandemic: levels, changes and validity of well-being data from Twitter. Evidence from ten countries.PLOS ONE

Dear Dr. Rossouw,

Thank you for submitting your manuscript to PLOS ONE. After careful consideration, we feel that it has merit but does not fully meet PLOS ONE’s publication criteria as it currently stands. Therefore, we invite you to submit a revised version of the manuscript that addresses the points raised during the review process. Your revised manuscript has been assessed by an additional reviewer. We ask you to incorporate his comments (see document attached).

We look forward to receiving your revised manuscript.

Kind regards,

Florian Fischer

Academic Editor

PLOS ONE

Journal Requirements:

2. In your Methods section, please include additional information about your dataset and ensure that you have included a statement specifying whether the collection and analysis method complied with the terms and conditions for the source of the data.

4. Thank you for stating the following in the Acknowledgments Section of your manuscript: "We would like to thank the Luxembourg National Research Fund (FNR) for the generous funding of the project. Additionally, we thank the University of Johannesburg and Auckland University of Technology for contributing funds. We also thank AFSTEREO for the I.T. service provided."

Please remove any funding-related text from the manuscript and let us know how you would like to update your Funding Statement. Currently, your Funding Statement reads as follows: "FS FNR-14878312 Luxembourg National Research Fund https://www.fnr.lu/

TG N/A https://www.uj.ac.za/

SR N/A https://www.aut.ac.nz/

7. Please ensure that you refer to Figures 13, 14, 15, and 16 in your text as, if accepted, production will need this reference to link the reader to the figure.

8. Please include a copy of Table 13 which you refer to in your text on page 21.

Additional Editor Comments (if provided):

Reviewers' comments:

Reviewer's Responses to Questions

**Comments to the Author**

1. If the authors have adequately addressed your comments raised in a previous round of review and you feel that this manuscript is now acceptable for publication, you may indicate that here to bypass the “Comments to the Author” section, enter your conflict of interest statement in the “Confidential to Editor” section, and submit your "Accept" recommendation.

Reviewer #3: (No Response)

2. Is the manuscript technically sound, and do the data support the conclusions?

Reviewer #3: Partly

3. Has the statistical analysis been performed appropriately and rigorously? 

Reviewer #3: I Don't Know

4. Have the authors made all data underlying the findings in their manuscript fully available?

Reviewer #3: Yes

5. Is the manuscript presented in an intelligible fashion and written in standard English?

Reviewer #3: Yes

6. Review Comments to the Author

Reviewer #3: Please find my review, including suggestions and recommendations to improve the manuscript, in the file pone-d-21-37698r1_reviewer_1.pdf.

7. PLOS authors have the option to publish the peer review history of their article (what does this mean?). If published, this will include your full peer review and any attached files.

Reviewer #3: **Yes: **

---

## [Author Response · Author response to Decision Letter 1]

18 Aug 2022

Please see attached WORD document since it contains graphs and tables that cannot be pasted into this space.

Kind regards

---

## [Decision Letter · Decision Letter 2]

9 Sep 2022

A year of pandemic: levels, changes and validity of well-being data from Twitter. Evidence from ten countries.

PONE-D-21-37698R2

Dear Dr. Rossouw,

We’re pleased to inform you that your manuscript has been judged scientifically suitable for publication and will be formally accepted for publication once it meets all outstanding technical requirements.

Kind regards,

Florian Fischer

Academic Editor

PLOS ONE

Reviewers' comments:

Reviewer's Responses to Questions

**Comments to the Author**

1. If the authors have adequately addressed your comments raised in a previous round of review and you feel that this manuscript is now acceptable for publication, you may indicate that here to bypass the “Comments to the Author” section, enter your conflict of interest statement in the “Confidential to Editor” section, and submit your "Accept" recommendation.

Reviewer #3: All comments have been addressed

2. Is the manuscript technically sound, and do the data support the conclusions?

Reviewer #3: Yes

3. Has the statistical analysis been performed appropriately and rigorously? 

Reviewer #3: Yes

4. Have the authors made all data underlying the findings in their manuscript fully available?

Reviewer #3: Yes

5. Is the manuscript presented in an intelligible fashion and written in standard English?

Reviewer #3: Yes

6. Review Comments to the Author

Reviewer #3: The authors addressed all the points I raised during review and also added comprehensive robustness analyses and a discussion sections, which improve the manuscript significantly. Where the manuscript tend to be 'over-conclusive', the claims made in the manuscript were attenuated and contextualized. I therefore recommend publishing the manuscript in PLOS ONE.

7. PLOS authors have the option to publish the peer review history of their article (what does this mean?). If published, this will include your full peer review and any attached files.

Reviewer #3: **Yes: **Julian Kauk

---

## [Editor Report · Acceptance letter]

27 Sep 2022

PONE-D-21-37698R2 

A year of pandemic: levels, changes and validity of well-being data from Twitter. Evidence from ten countries 

Dear Dr. Rossouw:

I'm pleased to inform you that your manuscript has been deemed suitable for publication in PLOS ONE. Congratulations! Your manuscript is now with our production department. 

Kind regards, 

on behalf of

Dr. Florian Fischer 

Academic Editor

PLOS ONE